# Nonsense mutations can increase mRNA levels

Precious O. Owuamalam*, Md. Nazmul Hossain and Saverio Brogna‡

**ABSTRACT**

Nonsense mutations can reduce mRNA levels, as premature translation termination may lead to the activation of nonsense-mediated mRNA decay (NMD). To examine how positional context influences these outcomes, we introduced premature translation termination codons (PTCs) at 15 locations within the coding region of a GFP reporter gene in *Schizosaccharomyces pombe*. PTCs in the first third of the coding region consistently led to reduced mRNA levels. In contrast, most downstream PTCs showed modest or minimal reductions, and several were associated with increased mRNA levels relative to the PTC-less control transcript. Measurement of transcript stability for one such variant indicated that the increased abundance was not attributable to decreased turnover. *UPF1* deletion in wild-type cells elevated the levels of transcripts that were reduced and, unexpectedly, further increased transcripts' abundance to exceed the control level. In spliced versions of these constructs, downstream PTCs generally reduced mRNA levels regardless of exon junction position. Overall, these observations indicate that an unexpected consequence of nonsense mutations can be increased mRNA levels. These findings may aid in the interpretation of the effects of nonsense mutations on mRNA abundance beyond the predictions of current NMD models and may also help in the design of eukaryotic gene expression constructs.

KEY WORDS: NMD, Splicing, mRNA surveillance, UPF1, GFP

## INTRODUCTION

Nonsense-mediated mRNA decay (NMD) is considered a quality-control mechanism of gene expression in eukaryotic cells that detects and eliminates mRNAs carrying a premature translation termination codon (PTC). The central function of NMD is therefore thought to be the elimination or the limitation of the accumulation of a broadly defined category of aberrant transcripts, which, if translated, would result in the wasteful production of truncated and potentially toxic polypeptides (Kervestin and Jacobson, 2012; He and Jacobson, 2015; Kurosaki and Maquat, 2016; Brogna and Wen, 2009; Fatscher et al., 2015). NMD is also believed to regulate the expression of normal transcripts, such as those containing short upstream open reading frames (uORFs) or stop codons in an abnormal sequence context (Tan and Wilkinson, 2025). This includes splice variants with poisonous exons encoding a PTC that functions to repress gene expression (Neil

School of Biosciences and Birmingham Centre for Genome Biology (BCGB), University of Birmingham, Edgbaston B15 2TT, UK.
*Present address: Wellcome-Wolfson Institute for Experimental Medicine, Queen's University Belfast, Belfast BT9 7BL, UK.

‡Author for correspondence (s.brogna@bham.ac.uk)

 S.B., 0000-0001-7063-4381

et al., 2025). NMD might also regulate cryptic noncoding RNAs and long noncoding RNAs (lncRNAs) because these transcripts often contain short open reading frames (Singh, 2024; Malabat et al., 2015). Whether NMD acts primarily as an mRNA surveillance system or as a broad gene-regulatory mechanism remains unclear, and its physiological significance is still debated across organisms (Brogna and Wen, 2009; Brogna et al., 2016). Despite these uncertainties, NMD is of considerable medical importance. Approximately 30% of human inherited genetic diseases are caused by nonsense or frameshift mutations that introduce a PTC (Miller and Pearce, 2014). For such allelic variants, whether the resulting transcript is subject to NMD and the extent to which it is affected are important determinants of disease severity and key criteria in assessing the clinical relevance of newly identified nonsense and frameshift variants in diagnostic sequencing (Coban-Akdemir et al., 2018). Additionally, whether a PTC-containing transcript is affected by NMD is relevant in cancer biology, particularly as a predictor of the efficacy of cancer immunotherapy (Litchfield et al., 2020).

The mechanism that distinguishes premature translation termination from termination at a normal stop codon is complex and arguably remains a major unresolved question (Brogna et al., 2016; Brogna and Wen, 2009). Many different NMD models have been proposed both within and across organisms. Two broad conceptual models have been proposed. The first is the 'faux 3′ UTR' model, first proposed in *Saccharomyces cerevisiae*, which posits that the key determinant of an NMD-inducing PTC is an abnormally long 3′UTR (Schweingruber et al., 2013; Amrani et al., 2004). The other is the exon junction complex (EJC) model, first proposed in mammalian cells, which proposes that a nonsense mutation induces apparent NMD only when located sufficiently upstream of at least one splice junction (Kurosaki and Maquat, 2016). The key assumption of both models is that the signal that distinguishes a normal from an abnormal stop codon and induces activation of the NMD pathway is the presence of a feature downstream of the termination codon: an abnormally long 3′UTR or the presence of an EJC (Muñoz et al., 2023) or the increased accumulation of NMD-inducing factors, such as UPF1 (Hogg and Goff, 2010). While these models can be applied as rough predictors of whether a given PTC-containing transcript will be affected by NMD, there are many observations that neither model can explain. Additionally, in many instances, nonsense mutations predicted to be subject to NMD have little or no effect on the mRNA level (Fang et al., 2013; Brogna and Wen, 2009; Brogna et al., 2016).

Here, we examined the effects of 15 nonsense mutations distributed throughout a GFP reporter gene on mRNA levels in *Schizosaccharomyces pombe*. GFP reporter genes have been well characterised in the context of NMD in several organisms, including *S. pombe*, a good model organism for studying NMD and testing current models (Wen and Brogna, 2010; Mendell et al., 2000). Results obtained with this reporter gene have been shown to mirror those obtained with endogenous genes (Wen and Brogna, 2010; Mendell et al., 2000). Unlike *S. cerevisiae*, approximately 50% of its genes contain introns, and there is evidence that nonsense

Biology Open

mutation-induced mRNA reduction often depends on splicing, but this dependence cannot be explained by an EJC-like mechanism as described for mammalian cells (Wen and Brogna, 2010). The data unexpectedly show that the GFP coding region is largely refractory to NMD; in most cases, the presence of a nonsense mutation does not induce a reduction in mRNA level. In fact, in many instances, the mRNA level may even increase compared to the wild-type (WT) control, and for a PTC located after the protein functional domain, it also leads to an increase in GFP fluorescence. We discuss how these data might challenge and refine current assumptions about NMD and how they may also be informative when designing expression constructs.

## RESULTS
### Most of the GFP coding region is refractory to NMD and, at some positions, a PTC leads to increased mRNA levels
In order to extend our previous analysis of nonsense mutations on mRNA levels, we generated additional constructs at GFP codon positions 1, 40, 53, 70, 88, 108, 112, 126, 141, 161, 185, 210, and 231, totalling 15 PTCs, including PTC6 and PTC27 from our previous study (Fig. 1A) (Wen and Brogna, 2010). We integrated these constructs (where GFP is under the control of the *nmt41* promoter and *adh1* terminator) into the *leu1* chromosomal locus in a WT *S. pombe* strain to minimise cell-to-cell variation due to differences in plasmid copy number, as previously described (Wen and Brogna, 2010). The constructs were extensively characterised and sequenced (Fig. S1A–D). We observed that mutations in positions 1–88 cause an apparent decrease in mRNA levels (Fig. 1B). However, for PTCs located downstream from PTC108 (except for PTC126), there was no change in mRNA levels. Surprisingly, three of the mutations (PTC141, PTC185, and PTC231) led to increased mRNA levels. We designated Region 1 as comprising the PTCs early in the coding region, which caused reduced mRNA levels (PTC1–PTC88), and Region 2 (PTC108–PTC231) as comprising mutations located further downstream, which cause no reduction in mRNA levels or, in some cases, cause an increase in the mRNA levels (Fig. 1A,C). When we compared the mean fold changes between the two regions, we observed a significant difference, further reinforcing the sharp difference in NMD sensitivity between the two regions (Fig. 1C).

To examine whether the observed mRNA reduction depends on *UPF1*, we selected two representative mutations from the two regions, PTC6 (Region 1) and PTC141 (Region 2). Deletion of *UPF1* increased PTC6 mRNA levels but unexpectedly also increased PTC141 mRNA levels, which were already significantly higher than those of PTC-less GFP (WT-GFP) (Fig. 1D). We then examined the changes in RNA stability. PTC6 had lower mRNA stability (with a half-life of approximately 6 min) than WT-GFP (approximately 50 min), and deletion of *UPF1* only partially increased its stability (to approximately 25 min) (Fig. 1E). WT-GFP and Δ*upf1*-GFP were similarly stable. While the increase in GFP-PTC6 mRNA levels (Fig. 1D) can partially be explained by increased RNA stability, since the deletion of *UPF1* largely restored the steady-state mRNA levels to WT levels and increased its RNA stability, that of PTC141 cannot. Although we did not test RNA stability changes between GFP-PTC141 and Δ*upf1*-GFP-PTC141, we did not observe any significant changes in the stability of GFP-PTC141 (which produced more mRNA than the WT-GFP) compared to WT-GFP (Fig. 1F).

### PTC231 increases GFP expression
As part of the GFP reporter characterisation, we examined the expression levels of the different reporters. As expected, the

WT-GFP reporter produced detectable GFP fluorescence. In contrast, all of the PTC-containing reporters showed no detectable GFP signal, except for GFP-PTC231, which showed detectable fluorescence (Fig. S1C). Notably, its fluorescence level was approximately twice that of WT-GFP (Fig. S1D). A previous study using deletion and mutational analysis reported that the minimal domain necessary for GFP fluorescence lies between codons 7 and 229 (Li et al., 1997). Therefore, codon position 231 falls outside this minimal domain required for GFP fluorescence. However, it was nonetheless surprising that this reporter produced approximately twofold more GFP fluorescence than WT-GFP, given that the mRNA level was increased by only 30%.

### No evidence for a correlation between NMD resistance and translation re-initiation
Given that PTCs located in Region 2 showed little or no sensitivity to NMD, we explored potential mechanisms underlying this effect. Since translation re-initiation at a downstream ATG can prevent NMD activation (Sherlock et al., 2023), we considered whether such re-initiation occurs in our reporter constructs. The next methionine codon after the first ATG is located at codon position 79, nine codons away from PTC70 (Fig. S1A). Given that the level of the PTC70 transcript is increased relative to that of PTC53, it could be argued that this might be a consequence of translation re-initiation occurring at position 79. However, we considered this unlikely, since the level of the PTC70 transcript is still reduced to an extent similar to that of earlier PTCs. Additional methionine codons are located at positions 89, 154, 219, and 234 (Fig. S1A). When comparing transcript levels of PTCs upstream of these methionine codons with those further upstream or downstream, there is no correlation, indicating no evidence that any putative translation re-initiation mechanism noticeably impacts NMD in this system.

### Splicing alters the effect of nonsense mutations on mRNA levels
Previously, we showed that splicing enhances NMD in *S. pombe* (Wen and Brogna, 2010). To further explore how PTC mutations affect NMD depending on the intron's position, we created intron-containing versions of the reporters and integrated the pDUAL-GFPivs-PTC-containing reporters into the *leu1* locus in WT *S. pombe* (Fig. 2A). The insertion of an intron (at codon 110) enhances NMD, because when the intron is present, not only PTCs in Region 1 but also most of those located in Region 2 show reduced mRNA levels (Fig. 2B,C). The strongest effect is seen for the PTCs closest to the intron (PTC108ivs to PTC161ivs), while the PTCs further downstream, more distant from the intron and closer to the normal stop codon (PTC185ivs to PTC231ivs), are not reduced (Fig. 2B). However, we noted a few exceptions in which splicing appears to suppress NMD rather than enhance it; PTC1 and PTC53 showed increased mRNA levels in the GFPivs reporters compared with the intronless reporters (Fig. S2A).

The absence of a correlation between a putative translation re-initiation event and the extent to which the presence of a PTC reduces mRNA levels is also evident in the intron-containing constructs. For example, PTC70ivs shows lower mRNA levels than PTC53ivs, even though, as discussed above, it is theoretically more likely to undergo translation re-initiation due to the presence of an ATG codon at position 79. We examined whether the observed reductions in mRNA levels were linked to *UPF1*. In the absence of *UPF1*, we observed a recovery of mRNA levels in Region 1 (PTC40ivs). Unexpectedly, however, we observed an about twofold increase in PTC141ivs mRNA levels above the GFPivs-PTC-less mRNA (Fig. 2D), which is higher than the

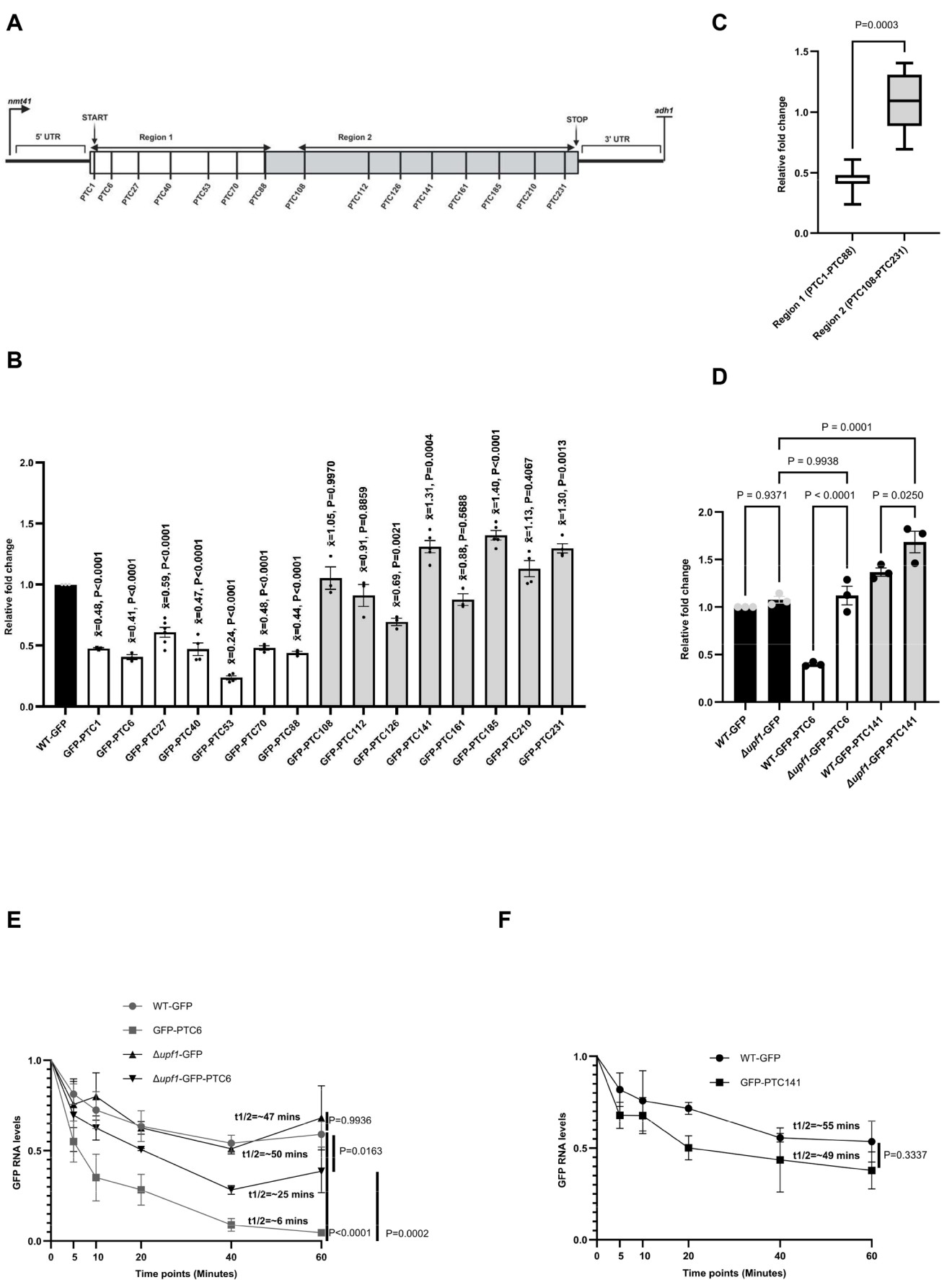

**Fig. 1.** See next page for legend.

**Fig. 1. Nonsense mutations can increase as well as reduce mRNA levels.** (A) Schematic of the pDUAL-GFP construct with the nonsense mutation positions. The codon sequence is split into two regions: Region 1, where mutations reduce mRNA levels; and Region 2, where mutations either increase or do not change mRNA levels. (B) Quantitative RT-PCR (qRT-PCR) comparing transcript levels of the GFP-PTC-containing transcripts. All data were normalised to WT-GFP RNA levels, with *rpl32* serving as an internal control. Error bars indicate standard errors from replicate experiments (at least three biological replicates were included in all cases, as represented by the data points). A one-way ANOVA with Dunnett's multiple comparisons was used to test for statistical significance ($\bar{x}$=mean fold change value). (C) Cumulative comparison of the mean fold change values of mutations in Region 1 (PTC1–PTC88) and mutations in Region 2 (PTC108–PTC231). Mann–Whitney rank-sum test was used to test for statistical significance. (D) qRT-PCR comparing transcript levels of WT-GFP, Δ*upf1*-GFP, GFP-PTC6, Δ*upf1*-GFP-PTC6, GFP-PTC141, and Δ*upf1*-GFP-PTC141. Transcript levels were normalised to *rpl32* as an internal control. Error bars represent the s.e.m. values between replicate experiments (*n*=3). A one-way ANOVA with Dunnett's multiple comparisons was used to test for statistical significance. (E) RNA stability quantifications of the WT-GFP, GFP-PTC6, Δ*upf1*-GFP, and Δ*upf1*-PTC6 transcripts after treatment with 1,10-phenanthroline (300 µg ml$^{-1}$) for the indicated time points. At least three independent biological replicates were included for each experiment. 18S rRNA was used as the internal control for qRT-PCR. For each transcript, each time point was normalised to time 0 (T0) in that transcript. Error bars represent the s.e.m. between replicate experiments. A two-way ANOVA with Dunnett's multiple comparisons was used to test for statistical significance. (F) RNA stability quantifications of the WT-GFP and GFP-PTC141 as in E. Error bars represent the s.e.m. between replicate experiments (*n*=3). A *t*-test was used to test for statistical significance.

1.5-fold increase seen in the intronless construct (Fig. 1D). We confirmed that our quantitative PCR (qPCR) primers were specific for the spliced transcripts and that we did not detect any corresponding pre-mRNA unspliced bands (Fig. S1B). Although we did not test the stability of PTC141ivs compared with WT-GFP directly, we observed an increase in mRNA levels in the GFPivs reporter compared with those of the intronless GFP reporter (Fig. 2E). However, the increased mRNA levels do not correlate with an increase in mRNA stability (Fig. 2F). The mechanism remains to be investigated; however, despite the reporters having the same promoter, there is more Pol II loading with the GFPivs reporter (Fig. 2G), suggesting that transcripts produced from intron-containing genes are transcribed more efficiently than those without introns. We also reconsidered whether the EJC is involved in the mechanism linking splicing to the nonsense-mediated changes in mRNA levels that we have described. In the human prototype case, the EJC is composed of a core of three proteins – Y14, MAGO, and FAL1 (EIF4A3) – as well as the peripheral subunit RNPS1. The EJC has not been extensively characterised in other species. Previously, we showed that the deletion of *RNPS1* or *MAGO* did not affect splicing-dependent NMD (Wen and Brogna, 2010). We here tested whether deleting *Y14* impacts this process (Fig. S2B). We could not examine the Δ*fal1* strain because the strain is severely growth impaired. As a control, we used the Δ*mago* strain containing the WT-GFPivs, PTC40ivs, and PTC141ivs plasmids. As in the Δ*mago* strain, we did not obtain any rescue with Δ*y14* (Fig. S2B,C).

Overall, the data show that splicing markedly alters the effects of nonsense mutations on mRNA levels regardless of whether the mutation is located upstream or downstream of the intron. Unexpectedly, in some instances, spliced transcripts show less nonsense-mutation-dependent mRNA reduction than the unspliced counterpart.

## DISCUSSION

In this study, we carried out detailed mutagenesis experiments using GFP reporter constructs to examine how nonsense mutations within the GFP coding region affect mRNA levels in *S. pombe*. We found that introducing a nonsense mutation between codons 1 and 88 caused the expected reduction in mRNA levels. However, introducing a mutation between codons 108 and 231 resulted either in no reduction (apart from one instance, position 126, where a modest reduction was observed) or, surprisingly, in an increase in mRNA levels when the mutations were located closer to the natural stop codon (positions 185 to 231). It appears that the GFP coding region (239 codons in total) can be divided into two distinct regions. The first, approximately one-third of the coding sequence, is a region in which a premature translation termination event can induce a reduction in mRNA levels. The remaining two-thirds of the coding region showed no reduction in mRNA levels. This latter portion may itself be divided into two subregions: an initial part in which premature translation termination does not alter the mRNA level, and a more distal part in which premature translation termination leads to a small increase in mRNA level. These observations are not what would be expected from the current NMD models. The constructs we are discussing lacked an intron and, therefore, were expected to follow the faux 3′UTR or similarly based NMD models. According to these, a nonsense mutation – apart from those very close to the normal stop codon – should result in a reduced mRNA level (Schweingruber et al., 2013; Amrani et al., 2004). This reduction should be proportional to the distance from the 3′ end of the transcript and should be most apparent when the PTC is furthest from the end of the transcript. The observations confirm the previous report that lengthening the distance between the PTCs and the normal stop codon does not suppress NMD in this organism (Wen and Brogna, 2010). These data therefore suggest that the key determinant of whether the mRNA is affected is whether premature translation termination occurs within a certain window after translation has begun. Translation is expected to be slow in the initial portion of the coding region, as is the case for other genes previously investigated (Ingolia et al., 2009; Bicknell et al., 2024; Weinberg et al., 2016). It is therefore plausible that a PTC would have an effect only during this early stage, when translation has not yet reached its optimal elongation rate, whereas once elongation has reached its optimal phase, premature termination of translation no longer affects the fate of the mRNA.

Another notable observation is that inserting a stop codon at position 1 (PTC1), most likely destroying translation initiation at the ATG start codon, also resulted in a similar level of mRNA reduction. The mRNA reduction caused by PTC1 is not expected to be due to NMD, since NMD is thought to be triggered by a premature translation termination event; thus, in the absence of initiation, there would not be any termination event. It is nevertheless interesting to consider, in view of the similarity of the reduced mRNA phenotype, whether a common feature underlies both events. A shared characteristic might be that these mRNAs never underwent or did not permit the ribosome to reach its optimal rate of translational elongation. In other words, mutations that, either directly or indirectly, prevent the ribosome from reaching an optimal elongation stage render the mRNA intrinsically short-lived.

One question that arises is why a PTC located close to the normal stop codon shows an increased mRNA level in the intron-less versions of the construct. This is particularly striking given that, at least for the transcripts tested, this increase does not appear to be due to enhanced mRNA stability and may therefore reflect increased mRNA production. One possibility is that lengthening the distance between the stop codon and the construct polyadenylation signal can increase gene expression. This is consistent with a previous report that lengthening the distance between the normal stop codon and the polyadenylation site can increase the mRNA level of a

**A**

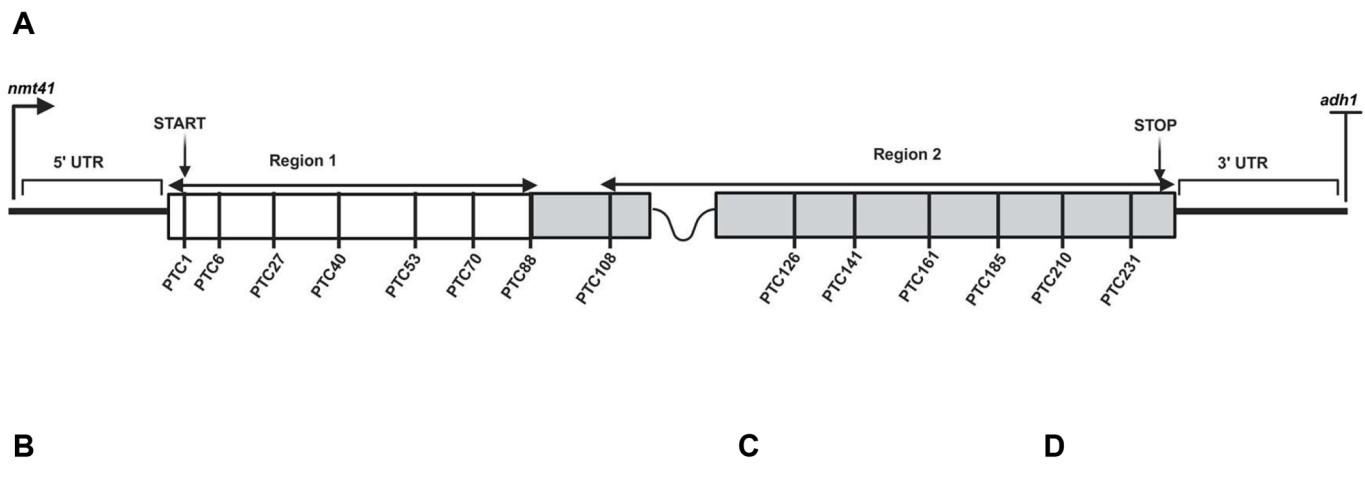

**B**

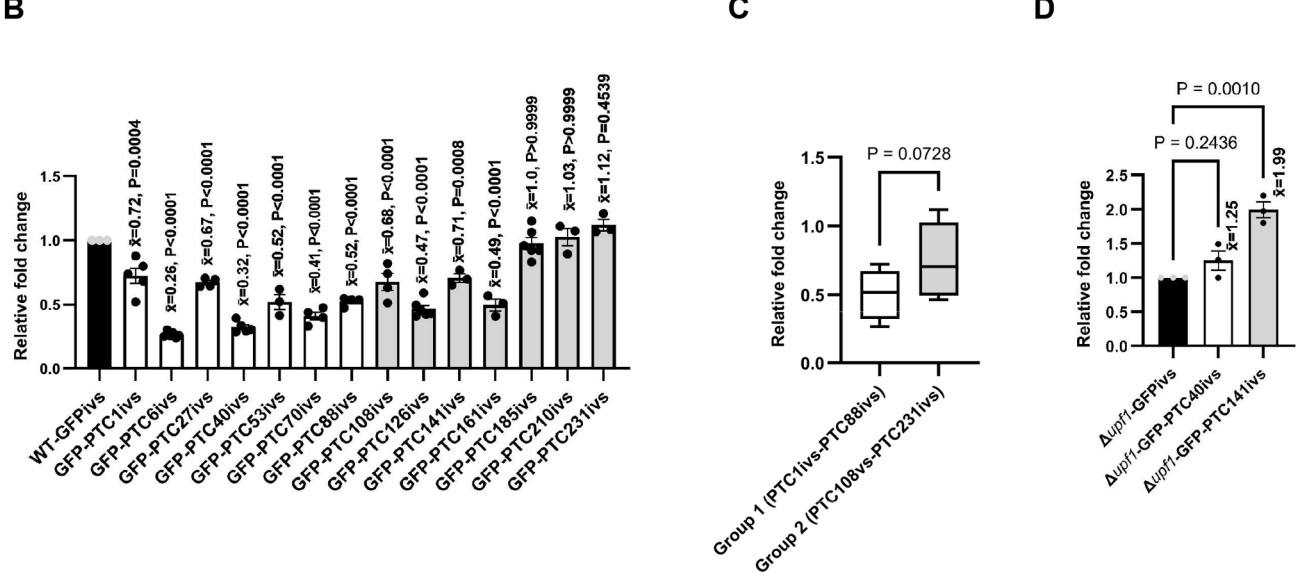

**C**

**D**

**E**

**F**

**G**

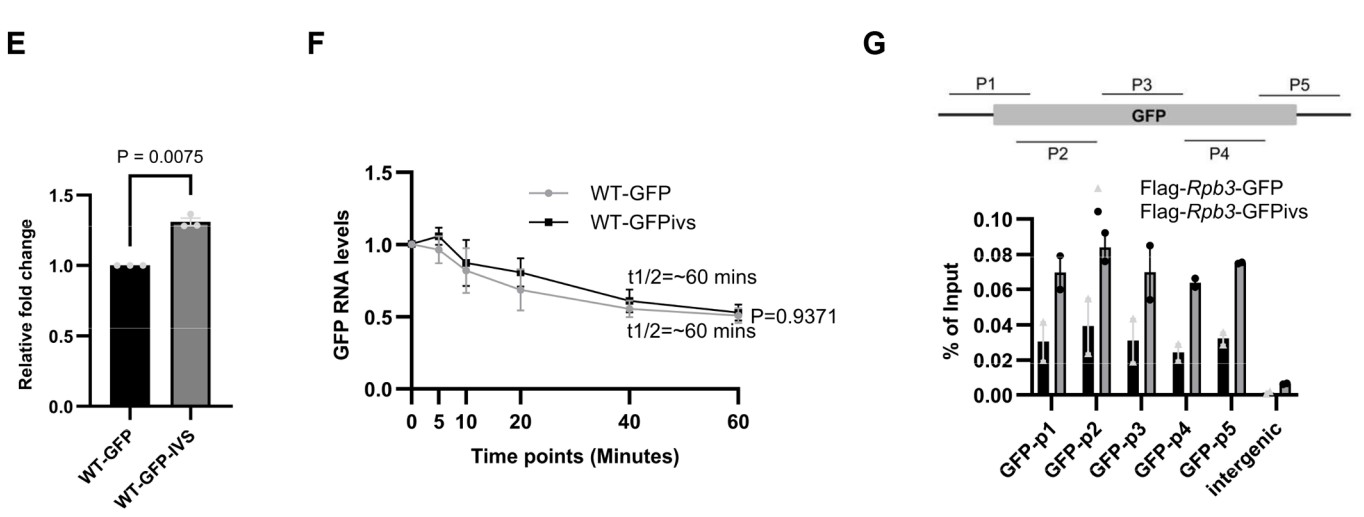

**Fig. 2.** See next page for legend.

**Fig. 2. Splicing alters the effect of a nonsense mutation on mRNA levels.** (A) Schematic of the intron-containing versions of the pDUAL-GFP constructs (pDUAL-GFPivs) with the nonsense mutation positions. The codon sequence is split into two regions: Region 1, where mutations reduce mRNA levels; and Region 2, where mutations either increase or do not change mRNA levels. (B) qRT-PCR comparing transcript levels of the GFP-PTCivs transcripts. All data were normalised to WT-GFPivs RNA levels, with *rpl32* serving as an internal control. Error bars indicate standard errors from replicate experiments (at least three biological replicates were included in all cases, as represented by the data points). A one-way ANOVA with Dunnett's multiple comparisons was used to test for statistical significance ($\bar{x}$=mean fold change value). (C) Cumulative comparison of the mean fold change values of mutations in Region 1 (PTC1ivs–PTC88ivs) and mutations in Region 2 (PTC108ivs–PTC231ivs). Mann–Whitney rank-sum test was used to test for statistical significance. (D) qRT-PCR comparing transcript levels of Δ*upf1*-GFPivs, Δ*upf1*-GFP-PTC40ivs, and Δ*upf1*-GFP-PTC141ivs. Transcript levels were normalised to Δ*upf1*-GFPivs, with *rpl32* serving as an internal control. Error bars represent the s.e.m. values between replicate experiments ($n$=3). A one-way ANOVA with Dunnett's multiple comparisons was used to test for statistical significance ($\bar{x}$=mean fold change value). (E) qRT-PCR comparing transcript levels of WT-GFP and WT-GFPivs. Transcript levels were normalised to WT-GFP, with *rpl32* serving as an internal control. Error bars represent the s.e.m. values between replicate experiments ($n$=3). A *t*-test was used to test for statistical significance. (F) RNA stability quantifications of the WT-GFP and WT-GFPivs after treatment with 1,10-phenanthroline (300 µg ml$^{-1}$) for the indicated time points. 18S rRNA was used as the internal control for qRT-PCR. For each transcript, each time point was normalised to time 0 (T0) in that transcript. Error bars represent the s.e.m. between replicate experiments ($n$=5). A *t*-test was used to test for statistical significance. (G) ChIP qPCR of Pol II ChIP signal (Flag-Rpb3) on WT-GFP and WT-GFPivs ($n$=2). The schematic above the bar chart illustrates the approximate positions of the ChIP qPCR primers on the GFP ORF. The primer sequences are listed in Table S2.

similar GFP reporter in *S. pombe* (Wen and Brogna, 2010). This suggests that the key determinant of gene expression that should be investigated is the distance between the stop codon and the cleavage and polyadenylation site, and possibly NMD, in light of a previous observation that nonsense mutations are linked to abnormal 3′ end cleavage and polyadenylation (Brogna, 1999). What is also notable is that a transcript displaying elevated mRNA levels shows an even higher increase in cells depleted of *UPF1*, and the effect seems to be more apparent with the spliced construct. The data suggest that this may not be a consequence of hyper-stabilisation of the transcripts but an increase in their production. Consistent with this interpretation, previous observations indicate that many transcripts which are upregulated in *UPF1* deletion strains are also shown to have increased Pol II loading in *S. pombe* (De et al., 2022). The mechanism is unknown, and it remains to be determined whether it might be linked to UPF1 presence at transcription sites, as reported in both *S. pombe* and *Drosophila* (Singh et al., 2019; De et al., 2022). Regardless of what the mechanism might be, this observation cautions against interpreting the upregulation of a given set of transcripts in UPF1-depleted cells as NMD targets that have been stabilised, as previously discussed (De et al., 2022).

Similarly to what has been reported previously in *S. pombe* and many other organisms, splicing radically increases the likelihood of a nonsense mutation leading to mRNA reduction. These data show that splicing enhances the mRNA reduction regardless of whether the intron is located upstream or downstream of the PTC, contrasting with the expectation of an EJC-like model, consistent with the earlier report in *S. pombe* (Wen and Brogna, 2010). This more extensive analysis indicates that, contrary to our earlier conclusion, there does not appear to be a linear correlation between PTC distance from the intron position and mRNA reduction. What the alternative mechanism might be remains unknown; some

hypothetical models have been considered to explain similar observations in *S. cerevisiae* (Wen et al., 2020 preprint). Consistent with previous observations that splicing enhances gene expression (Juneau et al., 2006; Le Hir et al., 2003), we noticed that the presence of an intron enhances the mRNA expression levels. However, this does not seem to be due to stability, which remains unchanged in our case. Instead, it is likely driven by an unidentified mechanism. Surprisingly, in some instances, splicing can also have the opposite effect of suppressing the NMD phenotype observed with the intronless construct (for example, PTC53 versus PTC53ivs and PTC1 versus PTC1ivs) (Fig. S2A).

In summary, contrary to a prediction of the view that cells have an effective mechanism that detects PTC-containing mRNAs as aberrant transcripts and destroys them by a specialised mechanism, we found that nonsense mutations often leave the mRNA level unchanged and, in some instances, can even increase mRNA levels without altering their stability. This implies that nonsense mutations can potentially increase mRNA production, perhaps by incidentally placing the stop codon at a more optimal distance from the polyadenylation site in the reporter gene. As for the reduction in mRNA level seen with the nonsense mutations in the early portion of the coding region, this may be primarily triggered by the ribosome detaching from the mRNA before it has reached the optimal elongation stage, and it is the absence of having achieved optimal translation speed that signals for mRNA reduction. The proposed model is that NMD may primarily be a consequence of a failure to establish an optimal translation circuit on the mRNA. Conceptually, this would be similar to the previously proposed ribosome release model of NMD (Brogna et al., 2016; Brogna and Wen, 2009), but with the emphasis that an optimally translated mRNA is not necessarily the one with the highest ribosomal load, as recently discussed (Bicknell et al., 2024). Notwithstanding the limitations of this study, as it is based on observations from a single reporter gene and a single organism, the conclusions should be relevant when designing eukaryotic expression constructs and when interpreting the phenotypic consequences of nonsense and frameshift mutations more broadly.

## MATERIALS AND METHODS
### Yeast strains
The list of *S. pombe* strains used in this study is shown in Table S1.

### Plasmid construction and integration
The plasmids used in this study were derived from the pDUAL vector (Matsuyama et al., 2004). All plasmid manipulations were performed on the pDUAL-GFP plasmid (Wen and Brogna, 2010). The different PTC mutations at codon positions 1, 40, 53, 88, 108, 112, 126, 161, 185, 210, and 231 were introduced into the pDUAL-GFP plasmid using site-directed mutagenesis (Liu and Naismith, 2008; Zheng et al., 2004) with some modifications. We used the Q5 DNA polymerase enzyme (New England Biolabs), along with primer pairs that have non-overlapping sequences at their 3′ end and complementary, overlapping sequences at the 5′ end. The PTC mutations were placed in the overlapping complementary sequences. We chose a minimum of 6 bases upstream and downstream of the desired codon mutations to correspond to the primer's complementary (overlapping) regions, with the primers terminating in a guanine or a cytosine. Our mutagenesis primers were about 50–60 bases in length, and the minimum melting temperature ($T_m$) difference between the non-overlapping and overlapping regions was 3°C. The non-overlapping regions were designed to be longer and of higher $T_m$ than the overlapping regions. Using these parameters, a two-stage PCR protocol was performed in a single run for each mutation. The first stage involved a standard PCR cycle: initial denaturation at 98°C for 30 s (one cycle), followed by 30–40 cycles of denaturation at 98°C for 10 s, annealing at a temperature dependent on the non-overlapping sequence for 30 s, extension at 72°C for 30 s per kilobase of plasmid and a final extension at 72°C for

Biology Open

10 min. The second stage included an optional denaturation at 98°C for 10 s, followed by one to three cycles of annealing at a temperature dependent on the overlapping sequence, and a final extension at 72°C for 10 min. All mutations were confirmed by Sanger sequencing. The amplified plasmids were then treated with DpnI (New England Biolabs) to destroy the methylated template plasmid before integration. To generate the intron-containing versions of the plasmids, the pDUAL-GFP plasmids with the nonsense mutations were digested with *Pml*1 restriction endonuclease (New England Biolabs), and the second intron of *ubc4* was amplified from *S. pombe* genomic DNA and cloned into the cut site at codon 110 using the NEBuilder HiFi DNA assembly kit (New England Biolabs) following the manufacturer's instructions. To generate the NMD reporter strains, the plasmids were integrated into the *leu1* locus as previously described (Wen and Brogna, 2010; Matsuyama et al., 2004) in the WT, Δ*upf1*, Δ*mago*, Δ*y14*, and the Flag-Rpb3-tagged strains. The list of primers used in this study is shown in Table S2.

### RNA extraction, analysis, and RNA stability quantification

Total RNA was extracted using the hot acid-phenol method (Collart and Oliviero, 2001). After extraction, the total RNA was treated with DNase I (1 unit) (Thermo Fisher Scientific) according to the manufacturer's instructions. First-strand cDNA synthesis was performed with the FastGene Scriptase II cDNA synthesis kit (Nippon Genetics) using 50 ng of total RNA, according to the manufacturer's instructions. Quantitative real-time PCR assays were conducted in 96-well plates (Applied Biosystems) using the ABI Prism™ SDS 7000 real-time PCR thermocycler (Applied Biosystems), following the manufacturer's instructions. PCR assays were performed using the qPCRBIO SyGreen Blue Mix Hi-ROX (PCR Biosystems). The $2^{-\Delta\Delta CT}$ method was used to calculate the relative expression levels of the target transcripts, normalised to *rpl32* mRNA or 18S rRNA. To inhibit transcription, cells were cultured to $OD_{600}$ (~0.7) in PMG media and treated with 300 µg ml$^{-1}$ 1,10-phenanthroline (Sigma-Aldrich). While this drug has been extensively used to inhibit transcription in yeast, including our previous NMD study in *S. pombe* (Wen and Brogna, 2010), its mechanism of action is not yet fully understood. There is evidence suggesting that the drug acts as a specific inhibitor of SUMOylation (McNeil et al., 2024); therefore, some of the observed inhibition could, in principle, reflect inhibition of SUMOylation. At different time points (0, 5, 10, 20, 40, and 60 min) immediately after adding the drug, 10 ml aliquots were transferred into 50 ml Falcon tubes (Sigma-Aldrich) containing about one volume of crushed ice before RNA extraction. mRNA decay rates were determined by fitting an exponential decay model to transcript abundance over time using the following formula:

$(\ln(y) = \ln(A) - kt)$.

For each time point, mRNA levels were first normalised to the 0-time point value. The natural logarithm of the normalised values was then plotted against time, and a linear regression was performed, corresponding to the first-order decay model. The decay constant ($k$) was obtained from the slope of the regression line, and the mRNA half-life was calculated as follows:

$t_{1/2} = \frac{\ln 2}{k}$.

### Chromatin immunoprecipitation (ChIP)

Freshly harvested cells from exponentially growing cultures ($OD_{600}$=0.5) were fixed for 5 min at room temperature with 1% formaldehyde (Sigma-Aldrich), followed by a 10-min incubation with a further addition of glycine (Sigma-Aldrich) to stop the cross-linking. ChIP was then conducted as previously described (De et al., 2022). The cell pellet was collected and washed three times with ice-chilled 1× PBS (Sigma-Aldrich), spinning at 5000 rpm for 5 min each. The supernatant was discarded, and the pellet was resuspended in ice-cold FA lysis buffer (100 mM HEPES-KOH, pH 7.5, 300 mM NaCl, 2 mM EDTA, 2% Triton X-100, 0.2% sodium deoxycholate) (Sigma-Aldrich) containing 1× protease inhibitor (EDTA-free protease inhibitor cocktail tablet, Roche). Cells were pelleted at 6000 rpm for 3 min at 4°C, and the pellet was resuspended in FA lysis buffer and zirconia beads (0.7 mm diameter, BioSpec). Cells were broken using a cell homogeniser (Bertin Instruments, Precellys 24, 12 cycles: 30 s at 5500 rpm and 2 min in ice). The bottom of each screw-cap tube was pierced three times with a red-hot 25G needle (BD Microlance), and each tube was immediately transferred to the barrel of a syringe fitted in a 15 ml Falcon tube (Sigma-Aldrich). The lysate was collected at 1000 rpm for 1 min at

4°C. To increase sonication efficiency and inhibit protease activity, 20 µl of 10% SDS and 20 µl of 100 mM PMSF (Sigma-Aldrich) were added to the mixture. Samples were sonicated for 16 cycles using a Bioruptor (Diagenode) to generate an average fragment size of 300–800 bp. The samples were spun at 13,200 rpm for 30 min at 4°C. The supernatant was collected and transferred to a fresh 1.5 ml DNA low-bind tube (Eppendorf). Fifty microlitres of Dynabeads Protein G (Thermo Fisher Scientific) were transferred to a 1.5 ml DNA low-binding tube and washed three times with 1 ml of 1× PBS containing 0.1% Tween 20 (Sigma-Aldrich). Ten microgrammes of antibody (Anti-Flag M2, Merck Millipore) were added, and the mixture was gently resuspended. The mixture was incubated on a rotor at room temperature for 10 min with gentle rotation. To remove unbound antibodies, the beads were washed three times with 1 ml of 1× PBS containing 0.1% Tween 20 (Sigma-Aldrich). After the final wash, the supernatant was removed, and antibody-coated beads were mixed with the appropriate amount of sonicated chromatin (300–400 µl) and transferred to a fresh 1.5 ml DNA-low-binding tube. Antibody-coated beads and the chromatin were then incubated overnight at 4°C on a rotor. The next morning, the supernatant was removed, and the beads were washed for 5 min at room temperature on a rotor as follows: two times with Wash Buffer I (50 mM HEPES-KOH pH 7.5, 150 mM NaCl, 1 mM EDTA pH 8.0, 1% Triton X-100, 0.1% sodium deoxycholate, 0.1% SDS), two times with Wash Buffer II (50 mM HEPES-KOH pH 7.5, 500 mM NaCl, 1 mM EDTA pH 8.0, 1% Triton X-100, 0.1% sodium deoxycholate, 0.1% SDS), two times with Wash Buffer III (10 Tris-HCl pH 8.0, 1 mM EDTA pH 8.0, 250 mM LiCl, 0.5% IGEPAL CA, 1% sodium deoxycholate), and two times with TE (10 mM Tris-HCl pH 8.0, 1 mM EDTA pH 8.0). After the final wash with TE, the beads were resuspended in 100 µl of elution buffer (EB) (50 mM Tris-HCl, pH 7.5, 10 mM EDTA, 1% SDS) and incubated for 10 min at 65°C, with occasional vortexing. The supernatant (eluate) was recovered and transferred to a fresh 1.5 ml DNA low-binding tube. Fifty microlitres of input DNA from the respective ChIP samples were taken, and 150 µl of EB was added to each to make a final volume of 200 µl. Both the ChIP material and the input were incubated at 65°C overnight to allow cross-linking. To remove proteins from DNA, 5 µl of proteinase K (20 mg ml$^{-1}$, Sigma-Aldrich) was added to both ChIP and input samples, and the samples were incubated at 50°C for 2 h. DNA was extracted using the Monarch PCR Purification Kit (New England Biolabs) according to the manufacturer's protocol. The qPCR reactions were prepared in triplicate using a 96-well plate (Applied Biosystems) and a Bio-Rad CFX Duet Real-Time PCR System. Each reaction contained 5 µl of 2× SyGreen Blue Mix Hi-ROX (PCR Biosystems), 2 µl of diluted input/IP, 1.5 µl of the 2.6 µM forward primer, and 1.5 µl of the 2.6 µM reverse primer to run 10 µl of reaction mixture in each well. The qPCR was performed with the following settings: 3 min at 95°C, followed by 40 cycles of 10 s at 95°C and 30 s at 60°C. Then the percentage of input was calculated using the following formulae:

adjusted input=input (CT)−log2×(input dilution factor);

% input=100×2$^{(\text{adjusted\_input CT−IP CT})}$.

### Microscopy and image quantification

Freshly harvested cells from exponentially growing cultures ($OD_{600}$=0.5) were used. One millilitre of the culture was briefly centrifuged to pellet the cells. The supernatant was discarded, and the pellet was washed in 1 ml of sterile water. After a brief second centrifugation to pellet the cells, the supernatant was discarded again, and the pellet was resuspended in sterile water. A 5 µl aliquot of the cell suspension was transferred to a slide with a coverslip (SLS), prepared for viewing under a Nikon Eclipse Ti fluorescence microscope. DIC images were captured with an exposure time of 100 ms, while GFP fluorescence (FITC) was visualised with an exposure time of 5 s using immersion oil objectives. The longer exposure time compared to typical GFP imaging reflects the expression system used here. The reporter construct is genomically integrated rather than episomally expressed, resulting in lower overall fluorescence intensity. Consequently, a longer acquisition time was required to reliably detect the signal without altering the gain or introducing excessive noise. Image quantification was performed with ImageJ, and total cell fluorescence (CTCF) was calculated using the following formula:

CTCF=integrated density−(area of the selected cell×mean background fluorescence).

## Statistical analysis

Statistical analysis was performed using GraphPad Prism™ (Version 10.6.1), which was also utilised to generate the figures. A *t*-test was employed to compare significant differences between the control and experimental groups when the data (of not more than two groups) were normally distributed. When the data were not normally distributed, the Mann–Whitney rank-sum test was used. For more than two data groups, ANOVA statistics were performed, followed by Dunnett's multiple comparisons. Statistical significance is indicated with absolute *P*-values.

## Acknowledgements

We thank Hannah Dixon for critically reading the first draft of the manuscript. We are also very grateful to the members of the laboratory for their valuable discussions and insightful feedback on the data underpinning this work.

## Competing interests

The authors declare no competing or financial interests.

## Author contributions

Conceptualization: P.O.O., S.B.; Formal analysis: P.O.O., M.N.H., S.B.; Funding acquisition: P.O.O., S.B.; Investigation: P.O.O., M.N.H., S.B.; Methodology: P.O.O., M.N.H., S.B.; Project administration: S.B.; Resources: S.B.; Supervision: S.B.; Validation: P.O.O., M.N.H., S.B.; Visualization: P.O.O., S.B.; Writing – original draft: P.O.O., S.B.; Writing – review & editing: P.O.O., S.B.

## Funding

The authors acknowledge funding from the Biotechnology and Biological Sciences Research Council [BB/M022757/1] (to S.B.), the Darwin Trust of Edinburgh (to P.O.O.), and the Bangabandhu Science and Technology Fellowship Trust (to M.N.H.). Open Access funding provided by University of Birmingham. Deposited in PMC for immediate release.

## Data and resource availability

All relevant data and details of resources can be found within the article and its supplementary information.

## First Person

This article has an associated First Person interview with the first author of the paper.

## Peer review history

The peer review history is available online at https://journals.biologists.com/bio/lookup/doi/10.1242/bio.062444.reviewer-comments.pdf

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
