## [Peer Review File · Biology Open]

Nonsense mutations can increase mRNA levels

Precious Obioha Owuamalam, Md Nazmul Hossain and Saverio Brogna
10.1242/bio.062444

Editor: Catherine L. Jackson

Review timeline

Original submission:	21 December 2025
Editorial decision:	12 January 2026
First revision received:	22 March 2026
Accepted:	24 March 2026

Original submission

First decision letter

MS ID#: bio.062444

MS Title: Nonsense mutations can increase mRNA levels

Authors: Precious Obioha Owuamalam, Md. Nazmul Hossain and Saverio Brogna

I have now reached a decision on the above manuscript.

The reviewer reports are shown at the bottom of this email.

As you will see, the reviewers gave favourable reports, but raised a few critical points that will require amendments to your manuscript. I hope that you will be able to carry these out, because we would like to be able to accept your paper.

At this stage, we also ask you to ensure your manuscript complies with our formatting guidelines - please see our manuscript preparation guidelines for details. Provided you are able to fully address the referees' comments, we are positive about publication of your paper (we accept over 95% of revision submissions) and therefore hope you won't mind any extra work involved in reformatting your manuscript at this point.

Please upload both a 'clean' version of your Word file, along with a highlighted version clearly showing where you have made changes in the revised manuscript. Please avoid using 'Track changes' in Word files as these are lost in PDF conversion.

I should be grateful if you would also provide a point-by-point response detailing how you have dealt with the points raised by the reviewers in the 'Response to Reviewers' box. Please attend to all of the reviewers' comments. If you do not agree with any of their criticisms or suggestions please explain clearly why this is so.

Reviewer 1

Comments for the author

Hossain et al. manuscript titled "Nonsense mutations can increase mRNA levels" investigates the consequences of introducing premature termination codons at various positions of a reporter GFP construct in the presence and absence of an intron in *S. pombe*. Their findings are highly intriguing

and challenge some assumptions in the nonsense-mediated decay (NMD) field. Therefore, the results of the manuscript are valuable and significant.

First, the authors show that PTCs affect mRNA abundance differently depending on their position along the mRNA. They divide the mRNA into two halves: PTCs in the first part, including PTC88, reduce mRNA levels, whereas PTCs between 108 and 231 largely maintain or increase mRNA levels relative to the control GFP construct. Despite maintaining similar or higher mRNA levels, the second-half PTCs, tested only by PTC141, can yield even higher mRNA abundance when the NMD pathway is mutated. The authors further show that, as expected, PTCs within the first half of the mRNA reduce mRNA stability, whereas second-half PTCs, demonstrated by PTC141, do not significantly change mRNA stability.

These findings are novel and interesting because the general assumption is that PTCs trigger NMD and reduce mRNA levels by directly affecting mRNA stability. However, the authors show that this is only true for PTCs closer to the start codon.

Secondly, the authors show that introducing an intron and thereby enabling pre-mRNA splicing can make some PTCs that previously showed no change in mRNA abundance now become sensitive to NMD. Therefore, for some PTCs that are farther from the start codon, pre-mRNA splicing plays a role in helping the NMD machinery detect these, but the authors show that this is unlikely to be facilitated by EJC.

Overall, the manuscript is very well written, concise, and clear. Figures are well prepared and contain sufficient detail to interpret them.

I have a few minor comments;

- 1- I recommend moving supplementary figure 1E to supplementary figure 2 to preserve consistency between figures and supplements.
- 2- The section regarding Y14 is somewhat unclear. Authors should explain their previous research on RNPS1 and MAGO and clarify their reasons for testing Y14.
- 3- The discussion should include a section on the location of the introduced intron. Could the position of the intron influence which PTCs become sensitive to NMD?
- 4- The ChIP method should include more specific details instead of just saying as previously described.

Reviewer 2

Comments for the author

Non-sense mediated decay (NMD) is a conserved eukaryotic mechanism that checks the presence of premature stop codons on mRNAs and induces their decay. The rules governing when an mRNA is detected by NMD are still unclear. The authors set out to use a GFP reporter system in *S. pombe* to test how positional context of the premature stop codons influences the mRNA level, hence possibly NMD.

The authors developed a gene, integrated in the genome, that encodes GFP with 15 versions having a premature termination codon (PTC). While adding a PTC up to codon 88 do result in less mRNA measured by qRT-PCR, the remaining ones (108-231, except 126) either have no influence or result in higher levels of mRNA. The reduction of PTC6 depended on NMD, as shown with the *upf1Δ* strain. Surprisingly, PTC141, which was already more abundant than WT-GFP, was even more abundant upon deletion of UPF1. Deleting UPF1 increased the mRNA stability of PTC6, but not of PTC141. Interestingly, GFP-PTC231 showed a higher fluorescence level (2x) than WT-GFP, much more than expected if there was a direct correlation between mRNA level and protein level. Finally, adding an intron in the sequence of GFP did reduce the mRNA level of most PTC mutants, except the last 3 ones. Again, surprisingly, mRNA levels of 2 PTC mutants were less reduced in the intron containing reporter than in the intronless reporter.

Overall, the work is solid and the approach of comparing many different PTC in the otherwise same transcript is interesting. The results obtained are quite surprising in several cases and add to our understanding of NMD. The conclusions are overall well supported by the data.

Comments:

1. One aspect for the mRNA stability results that need to be taken into account as a limitation is that 1,10-phenanthroline is more potent and quick at inhibiting SUMO than transcription (see PMID 38182817). Therefore, mRNA stability measurements may be a result of both SUMO inhibition and transcription inhibition. This should be at least briefly discussed.
2. Concerning the GFP fluorescence measurement:
 - The image presented seems saturated for PTC231 - is it true? In addition, I am very surprised by an acquisition time of 5s, this seems an order of magnitude too much.
 - The images were taken in water, which is an acute stress condition, removing the carbon source as well as changing osmotic parameters. Could it be that the GFP variant (PTC231) is in fact less sensitive to pH changes (for example) induced by the water stress, such that fluorescence is not a direct readout of GFP quantity but also of the excitation/emission capabilities of this variant? To confirm this result, a simple GFP western blot of WT and PTC231 should be performed.
3. The authors do not introduce the role of RNSP1, MAGO, Y14 or FAL1 and the rationale for the experiment presented in Fig S2 is not presented either.
4. Finally, a caveat of the experiments is that it is using an exogenous gene. It is unclear to me, since GFP fluorescence levels are not so useful, why the authors have not looked at an endogenous gene, which sequence has been evolutionarily selected in *S. pombe*. I would suggest that the use of GFP should be more motivated in the introduction.
5. There is a lack of consistency of the font size in figures.

Reviewer's Responses to Questions

Experimental quality

Does each figure have the proper controls?

If 'No', please indicate reasons in Comments for Author box below.

Reviewer #1:

- Yes

Reviewer #2:

- No

Were the data analyzed using appropriate statistical tests?

If 'No', please indicate reasons in Comments for Author box below.

Reviewer #1:

- Yes

Reviewer #2:

- Yes

Reproducibility

Were experiments performed using adequate number of biological replicates?

If 'No', please indicate reasons in Comments for Author box below.

Reviewer #1:

- Yes

Reviewer #2:

- Yes

Does the methods section provide sufficient detail to permit reproducibility?

If 'No', please indicate reasons in Comments for Author box below.

Reviewer #1:

- Yes

Reviewer #2:

- Yes

Completeness

Are the manuscript's conclusions supported by the data?

If 'No', please indicate reasons in Comments for Author box below.

Reviewer #1:

- Yes

Reviewer #2:

- Yes

Scholarship

Do the authors cite and discuss the merits of data that would argue for and against their conclusion?

If 'No', please indicate reasons in Comments for Author box below.

Reviewer #1:

- Yes

Reviewer #2:

- Yes

Does the manuscript title & abstract accurately reflect the contents of the manuscript, without hyperbole?

If 'No', please indicate reasons in Comments for Author box below.

Reviewer #1:

- Yes

Reviewer #2:

- Yes

First revision

Author response to reviewers' comments

We thank both reviewers for their positive evaluation and their helpful feedback on this manuscript. Our responses to the reviewers are provided in bold below each specific comment. Reviewer comments are reproduced in italics above each specific response.

Reviewer 1:

1- I recommend moving supplementary figure 1E to supplementary figure 2 to preserve consistency between figures and supplements.

Thank you for this feedback and the other comments below. We have moved Supplementary Figure 1E to a new Supplementary Figure 2 (Supplementary Fig. S2A).

2- The section regarding Y14 is somewhat unclear. Authors should explain their previous research on RNPS1 and MAGO and clarify their reasons for testing Y14 .

We have now provided additional background explaining why these experiments were performed in the Results section, where these experiments are described (Lines 161-165).

3- The discussion should include a section on the location of the introduced intron. Could the position of the intron influence which PTCs become sensitive to NMD?

In our earlier study ([10.1038/emboj.2010.48](https://doi.org/10.1038/emboj.2010.48)), we observed enhanced NMD regardless of the intron position within our reporter gene. Splicing enhanced NMD more when the intron was close to the PTC. Our data corroborate these findings, showing enhanced NMD whether a PTC is located upstream or downstream of the intron. However, our current, more extensive analysis indicates that, contrary to our earlier conclusion, there does not appear to be a linear correlation between PTC distance from the intron position and mRNA reduction. This has now been clarified in the Discussion (Lines 237-239).

4- *The ChIP method should include more specific details instead of just saying as previously described.*

In the revised manuscript, we have expanded the Methods section to give the details of our ChIP experiment (Lines 324-367).

Reviewer 2:

1. *One aspect for the mRNA stability results that need to be taken into account as a limitation is that 1,10-phenanthroline is more potent and quick at inhibiting SUMO than transcription (see PMID 38182817). Therefore, mRNA stability measurements may be a result of both SUMO inhibition and transcription inhibition. This should be at least briefly discussed.*

Thank you for this feedback and the other comments below. We now clarify in the revised manuscript that the mechanism by which 1,10-phenanthroline inhibits transcription is not fully understood and may, at least in part, reflect an indirect effect through inhibition of sumoylation. We also cite the recent paper that reported this recent observation (McNeil et al., 2024) (Lines 307-311).

2. *Concerning the GFP fluorescence measurement:*

-The image presented seems saturated for PTC231 - is it true? In addition, I am very surprised by an acquisition time of 5s, this seems an order of magnitude too much.

The PTC231 image is not saturated. Prior to data collection, we optimised the acquisition settings by testing different exposure times. An exposure time of 5 s provided the best signal-to-noise ratio while remaining within the detector's dynamic range, and pixel intensity histograms confirmed that saturation was not reached. We repeated imaging across multiple independent biological replicates (as reflected in the quantified data points in supplementary Figure 1D (Fig. S1D) and consistently observed the same fluorescence pattern for both GFP and PTC231 under these settings.

The longer exposure time compared to typical GFP imaging reflects the expression system used here. The reporter construct is genomically integrated rather than episomally expressed, resulting in lower overall fluorescence intensity. Consequently, a longer acquisition time was required to reliably detect the signal without altering the gain or introducing excessive noise. We have now clarified this in the Methods section to make the imaging parameters and rationale more transparent (Lines 376-379).

-The images were taken in water, which is an acute stress condition, removing the carbon source as well as changing osmotic parameters. Could it be that the GFP variant (PTC231) is in fact less sensitive to pH changes (for example) induced by the water stress, such that fluorescence is not a direct readout of GFP quantity but also of the excitation/emission capabilities of this variant? To confirm this result, a simple GFP western blot of WT and PTC231 should be performed.

We agree that cells resuspended in water are stressed and typically show weaker fluorescence. However, all our comparisons are between cells resuspended in water under the same conditions. It could be interesting to investigate whether the PTC231-produced protein is less sensitive to pH changes; however, we believe this would go beyond the scope of the present study. Similarly, adding a Western blot to assess whether PTC231 is translated more efficiently than the other constructs would be informative. However, this would not change the conclusions of this manuscript, and these experiments were not performed at the time and are now difficult to carry out.

3. *The authors do not introduce the role of RNSP1, MAGO, Y14 or FAL1 and the rationale for the experiment presented in Fig S2 is not presented either.*

We now provide background information explaining why these experiments were performed and have included this in the Results section, where the experiments are described. The

experiment was performed to further investigate whether the EJC plays a role in this mechanism, and Y14 had not been tested previously because, at the time, we were not certain that the protein was indeed the orthologue of human Y14. The results presented corroborate our earlier conclusion that the EJC does not play a role in the linkage between splicing and mRNA reduction (Lines 161-165).

4. Finally, a caveat of the experiments is that it is using an exogenous gene. It is unclear to me, since GFP fluorescence levels are not so useful, why the authors have not looked at an endogenous gene, which sequence has been evolutionarily selected in S. pombe. I would suggest that the use of GFP should be more motivated in the introduction.

In our previous study, we used an endogenous gene alongside GFP reporters. We have explained in the Introduction why GFP was chosen and clarified further in the Introduction section that the results obtained with this reporter gene mirror those obtained with endogenous genes (Lines 69-71). Regarding the question of whether endogenous genes in a given organism have been selected to contain features that make them more NMD-sensitive, and thus better regulated, we argue in the Discussion that this is unlikely to be the case, based on the variable effects that nonsense mutations have on endogenous genes across many systems, from yeast to humans. This is consistent with the view that eukaryotes may not have evolved a mechanism to survey mRNA in the manner envisaged by the classical NMD model (Lines 237-239).

5. There is a lack of consistency of the font size in figures.

We have made sure that the font sizes are similar in all figures.

Second decision letter

MS ID#: bio.062444R1

MS Title: Nonsense mutations can increase mRNA levels

Authors: Precious Obioha Owuamalam, Md. Nazmul Hossain and Saverio Brogna

I am happy to tell you that your manuscript has been accepted for publication in Biology Open, pending our standard publication integrity checks. It was accepted on 24th March 2026.